# The Alligator and the Mosquito: North American Crocodilians as Amplifiers of West Nile Virus in Changing Climates

**DOI:** 10.3390/microorganisms12091898

**Published:** 2024-09-14

**Authors:** Desiree Kirsten Andersen, Gracie Ann Fischer, Leigh Combrink

**Affiliations:** School of Natural Resources and the Environment, University of Arizona, Tucson, AZ 85721, USA; gfischer@arizona.edu

**Keywords:** West Nile Virus, crocodilian, mosquito, transmission, North America, One Health

## Abstract

In an age of emerging zoonoses, it is important to understand the intricate system of vectors and reservoirs, or hosts, and their relation to humans. West Nile Virus (WNV) has been detected in a myriad of nonhuman hosts. Transmission of the virus to humans is reliant on amplified seroprevalence within the host, which occurs primarily in birds. However, recent studies have found that other animal groups, including crocodilians, can obtain seroprevalence amplification to levels that make them competent hosts able to transmit WNV to mosquitoes, which can then transmit to humans. Climate change could exacerbate this transmission risk by shifting the distributions of mosquito vectors towards novel geographic ranges. Here, we use maximum entropy models to map the current and future distributions of three mosquito vector species and four crocodilian species in North America to determine the emerging risk of WNV outbreaks associated with changing climates and WNV associated with crocodilians in North America. From our models, we determined that one mosquito species in particular, *Culex quinquefasciatus*, will increase its distribution across the ranges of all crocodilian species in all tested climate change scenarios. This poses a potential risk to public health for people visiting and living near crocodilian farms and high-density natural crocodilian populations.

## 1. Introduction

The impact of climate change on the spread of vector-borne and zoonotic diseases is unknown, despite the significant potential effect on both wildlife and human health. The successful spread of vector-borne infectious disease pathogens is dependent on the spatio-temporal distribution of both their vectors [1] and hosts [2,3,4]. Climate change, coupled with anthropogenic land-use change and human population growth, could result in range shifts or expansions for many disease vectors [5], leading to potential novel vector–host associations and significantly altering disease transmission dynamics [6,7].

West Nile Virus (WNV) was first discovered in the New World in New York City in 1999 [8,9] and has since spread and become endemic to most of the continental United States [10]. The pathogen is a mosquito-borne arbovirus from the flavivirus group [11]. It is native to Europe, Africa, Asia, and Oceania [12] and is spread by the bite of infected *Culex* mosquitoes, primarily *Culex pipiens*, *Culex tarsalis*, and *Culex quinquefasciatus* [10,13]. Birds are the reservoir and amplifying hosts [12] with the disease being maintained through the mosquito–bird–mosquito transmission cycle [14]. However, the effect of the virus is dependent on species; some avian species are amplifiers and super spreaders (American robin, house finch and house sparrow), while others are super suppressors (e.g., Northern cardinals, Steller’s jays) having levels of viremia too low to transmit the virus to feeding mosquitoes [15]. Corvids (crows and ravens) are highly susceptible and succumb to the disease soon after infection [16]. Horses and humans are incidental or dead-end hosts [12,14], with many infections (70–80%) in humans being asymptomatic [12,17]. The incubation period in humans is 2–6 days [11,18] but can be up to 14 days [9], with symptoms including fever, fatigue, headache, muscle and joint pain, gastro-intestinal distress, and in some cases, a transient macular rash on the torso and extremities [19,20]. In <1% of infections, people can develop neuroinvasive diseases, such as meningitis, encephalitis, or paralysis [20,21].

American alligators (*Alligator mississippiensis*) are a common crocodilian species across the Southeastern United States. Natural densities of alligators range depending on local conditions, from 1.52 to 2.35 alligators/km in a wildlife area in east Texas [22] to around 2.56–9.02 alligators/km in Florida [23]. Alligator farms exist across their natural range, many of which can house large densities of these animals for display to the public (entertainment/tourism) or for farming (hides/meat/hunting). Recommended stocking rates for farmed alligators are around 10–20 alligators/acre; however, densities can be as high as 50/acre at some farming locations [24,25]. Alligator farms play key roles in the conservation of the species by discouraging poaching of wild alligators [26]. However, WNV has been found in such farmed alligators at amplified levels sufficient for transmission to humans via *Culex* mosquitoes [27,28], suggesting *A. mississippiensis* may play a role in the ecology of WNV as an amplifying host. Of the *Culex* species tested for propensity to feed on *A. mississippiensis*, feeding by *Cx. quinquefasciatus* was highest at 82.2%, followed by *Cx. pipiens* at 31.1%, and *Cx. tarsalis* at 6.7% [27]. In addition to causing lesions, which reduce the economic value of alligator hides [29], WNV can cause large, fatal outbreaks in juveniles [28]. Incidence of WNV is, however, not limited to American alligators; WNV antibodies have been found in wild and farmed Morelet’s, or Mexican crocodiles, *Crocodylus moreletii*, indicating the susceptibility of crocodilians as a group to WNV, although the species’ ability to amplify the virus and transmit it back to mosquitos has not been tested. Outside of North America, the Kunjin strain of WNV has been detected in farmed saltwater crocodiles in Australia, where it may be the cause of skin lesions [30]. Further studies in farmed saltwater crocodiles in Northern Australia show this strain is capable of mosquito-independent transmission, meaning the crocodile is capable of amplification and transmission through water via cloacal shedding [31].

In continental North America, there are four species of crocodilians: *Alligator mississippiensis* (American alligator), *Caiman crocodilus* (spectacled caiman), *Crocodylus acutus* (American crocodile), and *Crocodylus moreletii* (Morelet’s or Mexican crocodile) (Figure 1). To investigate the risk of WNV spillover from crocodilian species in North America, we used maximum entropy to model the overlap between *Culex* mosquito distributions and four species of crocodilians under different climate scenarios. Our data will help to identify potential hotspots of disease outbreaks and could inform future public health treatment and prevention initiatives targeting WNV.

## 2. Materials and Methods

To model mosquitoes and crocodilians in North America (Figure 2), we first obtained occurrence data from the Global Biodiversity Information Facility (GBIF [32,33]). We focused on three *Culex* species (*Culex pipiens*, *Culex quinquefasciatus*, and *Culex tarsalis*) and four North American crocodilian species (*Alligator mississippiensis*, *Caiman crocodilus*, *Crocodylus acutus*, and *Crocodylus moreletii*). GBIF data were restricted to continental North America and filtered to include human observations and specimens and to exclude records with geospatial issues. This resulted in 227,615 observations of *Cx. pipiens*, 48,778 observations of *Cx. quinquefasciatus*, 64,943 observations of *Cx. tarsalis*, 34,033 observations of *A. mississippiensis*, 1977 observations of *C. crocodilus*, 3499 observations of *C. acutus*, and 4897 observations of *C. moreletii*. To reduce spatial autocorrelation, occurrences of all species were spatially thinned to the resolution of the environmental layers (0.04166 decimal degrees, or ~4 km) via the removal of duplicate records function of our model algorithm. The final models were therefore trained with 369 occurrences of *Cx. pipiens*, 773 occurrences of *Cx. quinquefasciatus*, 680 occurrences of *Cx. tarsalis*, 1929 occurrences of *A. mississippiensis*, 158 occurrences of *C. crocodilus*, 135 occurrences of *C. acutus*, and 290 occurrences of *C. moreletii*.

Environmental layers included 19 bioclimatic variables from WorldClim 2.1 [34], monthly solar radiation, monthly water vapor pressure, monthly wind speed, and derived landcover variables (World Land Cover at 30 m resolution from MDAUS BaseVue 2013). In total, we had 66 environmental variables (Appendix A). The spatial resolution of these variables was 2.5 arc minutes, or 0.04166 decimal degrees (~4 km). To reduce the chance of overfitting via the inclusion of highly correlated variables, we ran initial MaxEnt (version 3.4.4) [35] models for each species using all variables and default settings utilizing the ‘dismo’ package [36] in R ver. 4.3.3. We next removed highly correlated variables (Pearson coefficient > 0.8), which had a lower contribution retaining variables with high contribution and low correlation. This resulted in each species being modeled with a different set of variables (Appendix A).

We then modeled habitat suitability across North America using selected variables and utilizing MaxEnt version 3.4.4, with model tuning conducted in R v. 4.1.3 using the ‘ENMeval’ package [37]. For model tuning, we used feature classes “L”, “LQ”, “LQH”, and “H” and regularization multipliers 0.5, 1, 2, 5, 10, and 20. Spatial bias was reduced during ENMevaluate runs, which removed duplicate occurrences within the same environmental layer cell. The resulting numbers of unique occurrences were thus reduced to 384 for *Cx. pipiens*, 805 for *Cx. quinquefasciatus*, 687 for *Cx. tarsalis*, 1981 for *A. mississippiensis*, 169 for *C. crocodilus*, 145 for *C. acutus*, and 301 for *C. moreletii*. True skill statistic (TSS [38]) was calculated for each model output created by ENMevaluate, and a weighted average output was created using the TSS value as the weighting factor. These models were projected to four climate change scenarios (SSP126—“sustainability”, SSP245—“middle of the road” or “regional rivalry”, SSP370—“inequality”, and SSP585—“fossil-fueled development”) [39] and two climate models of low and high climate sensitivity (MIROC6 [40] and CNRM-CM6-1 [41], respectively) across four time periods (2021–2040, 2041–2060, 2061–2080, and 2081–2100). *Culex* mosquito models were then thresholded by TSS values and intersected with each thresholded crocodilian model. *Culex* distributions were also overlapped with existing crocodilian farms (Appendix A).

## 3. Results

### 3.1. Model Fit Statistics

Our final models obtained fair to excellent fit statistics for both AUC (fair: 0.7–0.8; good: 0.8–0.9; excellent: >0.9) and TSS (fair: 0.4–0.5; good: 0.5–0.7; very good: 0.7–0.85; excellent or near perfect: 0.85–1). The model for *A. mississippiensis* had fair AUC (0.7495 ± 0.0029) and excellent TSS (0.9358 ± 0.0095); low AUC values are common for widespread species [42], but high TSS indicates an excellent ability of the model to predict presence vs. absence [38]. The *C. crocodilus* (AUC = 0.9612 ± 0.0008; TSS = 0.9903 ± 0.0063), *C. acutus* (AUC = 0.9626 ± 0.0033; TSS = 0.9786 ± 0.0121), and *C. moreletii* (AUC = 0.9296 ± 0.0045; TSS = 0.9606 ± 0.0095) models all had excellent model fit. For *Culex* models, *Cx. pipiens* had good AUC (0.8755 ± 0.0166) and very good TSS (0.7695 ± 0.0615); *Cx. quinquefasciatus* had good AUC (0.8301 ± 0.0064) and very good TSS (0.8397 ± 0.0130); and *Cx. tarsalis* had good AUC (0.8070 ± 0.0165) and good TSS (0.6505 ± 0.0514).

### 3.2. Variable Contribution

Our model for *A. mississippiensis* used a total of 14 variables (Appendix A), with July water vapor pressure (vapr) having the highest contribution (39.1%) and permutation importance (42.4%). Woody wetland percent and March vapr also had high contribution and permutation importance, respectively. For *C. crocodilus*, whose model used a total of 28 variables (Appendix A), December vapr had the highest contribution (60.7%), but Bio4 (isothermality; 27.4%) and March vapr (24.7%) had the highest permutation importance. For *C. acutus* (26 variables; Appendix A), the highest contribution and permutation importance was November vapr at 49.9% and 48.7%, respectively. Similarly, the *C. moreletii* model (24 variables; Appendix A) also had November vapr as the highest contribution (40.7%) and permutation importance (40.0%) with January vapr also having a notable contribution (36.2%).

Conversely to our crocodilian models, the *Culex* species models favored different variables more related to climate and landscape. For example, *Cx. pipiens*’ distribution was influenced mainly by the percent of urban area (contribution = 66.0%, permutation importance = 29.7%) out of 22 variables used (Appendix A). The *Cx. quinquefasciatus* model (18 variables; Appendix A) had January vapr as the highest contribution (46.1%) but Bio11 (mean temperature of coldest quarter) as the highest permutation importance (47.8%). Finally, the *Cx. tarsalis* model (18 variables; Appendix A) had the percent of urban area as the highest contribution (38.6%) and September solar radiation as the highest permutation importance (28.9%).

### 3.3. Area and Overlap Statistics

Currently, *Alligator mississippiensis* has a suitable area of 928,440 km^2^, which will experience a 1.9% decrease (CNRM 585 2100) to 0.5% increase (MIROC 585 2100). *Caiman crocodilus* currently has a suitable area of 220,883 km^2^ and will experience a 13.0% decrease (MIROC 585 2100) to a 13.0% increase (MIROC 245 2060). *Crocodylus acutus* currently has 289,439 km^2^ of suitable area, which will experience a 4.3% decrease (CNRM 370 2100) to a 0.5% increase (MIROC 245 2080). Finally, *C. moreletii*, which has a suitable area of 355,749 km^2^, will increase from 3.9% (MIROC 585 2040) to 28.1% (MIROC 585 2100).

For mosquitoes, *Cx. pipiens* has a current suitable area of 4020,285 km^2^, which will increase from 2.4% (MIROC 370 2040) to 21.4% (CNRM 585 2100). *Culex quinquefasciatus* at 3,297,324 km^2^ will increase from 4.9% (CNRM 126 2040) to 29.1% (MIROC 585 2100). Finally, *Cx. Tarsalis* with the largest suitable area of 5,697,260 km^2^ will experience a 1.3% increase (MIROC 585 2040) to an 8.6% increase (CNRM 585 2100).

The mosquito species currently with the highest overlap for all crocodilian species is *Culex quinquefasciatus* (Figure 3). It currently covers 77.0% of *A. mississippiensis’* range (Figure 4), 98.4% of *C. crocodilus*’ range, 98.5% of *C. acutus’* range, and 99.5% of *C. moreletii’s* range (Table 1). While this overlap is predicted to increase for *A. mississippiensis* anywhere from 5.7% (CNRM 126 2040) to 22.7% (MIROC 585 2100), the overlap of *Cx. quinquefasciatus* is not predicted to change at all for the other crocodilian species. Although *Cx. pipiens’* suitable range is expected to increase for all climate change scenarios, this is not reflected in overlap with any crocodilian species; this overlap is expected to decrease or not change at all. The overlap of *Cx. tarsalis* will also decrease across the ranges of *C. crocodilus*, *C. acutus*, and *C. moreletii* but will have variable change across the range of *A. mississippiensis*.

## 4. Discussion

### 4.1. Species-Specific Distribution Changes

We found species-related differences in *Culex* mosquito distribution under different climate scenarios. *Cx. quinquefasciatus*, the southern house mosquito, shared the highest percent overlap with all crocodilian species in North and Central America for all climate change scenarios (up to 99.9% for some species/scenarios). Despite this mosquito being considered a low-competency vector for WNV in crocodilian species [27,43], we suspect that, owing to this extensive overlap, *Cx. quinquefasciatus* will likely emerge as a factor in the increased risk of West Nile Virus for both American alligators and humans in close proximity. This species’ distribution is driven partially by the mean temperature of the coldest season and favoring warmer winter temperatures (Appendix A), which explains its expansion to the north and maintained distribution in the south, similar to a previous study predicting an increase in suitability across the species’ range in North and South America [44].

*Culex pipiens*, which showed moderate competence as a WNV vector for crocodilians [27], is predicted to move northward in its distribution for all climate change scenarios based on our models. As such, this will lead to a decreased overlap in occurrence for the mosquito with all North and Central American crocodilian species. This shift suggests that *Cx. pipiens* will not be an emerging threat for WNV transmission among crocodilians, as its distribution becomes increasingly disconnected from the habitats of these reptiles.

*Culex tarsalis*, identified as potentially the most competent WNV vector for crocodilians [27], shows the highest degree of change in overlap with these species. This vector’s distribution is highly sensitive to climatic changes [44], which could result in variable risk profiles for WNV transmission. Compared to a recent study [44], our models for *Cx. pipiens* and *Cx. quinquefasciatus* both show similar patterns of changes in suitability across North America, which is predicted to decrease in the south and increase in the north for both species.

### 4.2. Crocodilian Risk Profiles

Although we did find a predicted increase in crocodilian species’ suitable areas, they are less mobile than mosquitoes, so we do not expect their ranges to shift drastically. However, climate change and continued anthropogenic pressures, including habitat alteration such as pesticide use [45,46] and water management practices [47], are threats to crocodilians that will likely cause range decreases and limit range expansion.

Our crocodilian hosts exhibit divergent risk profiles for WNV infection and transmission based on their overlap with *Culex* mosquito species. For instance, *C. crocodilus* currently has the lowest overlap with any *Culex* species, with only 0.2% overlap with Cx. pipiens and 8.6% overlap with *Cx. tarsalis*. *Crocodylus acutus* and *C. moreletii* both face a similar risk from *Cx. quinquefasciatus* but have low and decreasing overlap with the other two *Culex* species. In contrast, *A. mississippiensis* experiences significant overlap with *Cx. quinquefasciatus*, currently at 77.0%, and projected to increase to 99.7% by 2100 under SSP 585 (MIROC). Overall, *A. mississippiensis* is likely the most at risk due to its range overlapping with all three *Culex* species.

### 4.3. Conservation and Public Health Implications

After being placed on the Endangered Species List in 1967, *A. mississippiensis* was bred in captivity to establish farms throughout the southeastern United States [48]. A combination of private-sector farming and governmental poaching regulations rapidly increased the population of *A. mississippiensis*, resulting in its removal from the Endangered Species List 20 years later. The establishment of these farms in the Southeastern United States also created a substantial industry for both hunting and entertainment tourism, with the sale of skins contributing USD 64.2 million to the economy of Louisiana in 2015 alone [24].

Farms with high alligator densities in areas with a projected increase in the overlap of WNV vector species can become vulnerable to die-offs of both farmed and wild crocodilian species. *Alligator mississippiensis* species inoculated with WNV have been experimentally shown to shed the virus through the cloaca [28]. This viral shedding was also shown to infect non-inoculated *A. mississippiensis*, with the transmission of the virus suspected to occur through a shared water environment. This hypothesized transmission route could increase the likelihood of infection in crocodilians that share a common water source. Large die-offs due to both vector and environmental transmission would create economic devastation for both entertainment/tourism and hide/meat/hunting farms, resulting in losses of hundreds of millions of dollars to the alligator farming industry.

Dense congregations of crocodilians at both public and industrial farms in areas with projected vector species overlap may increase the risk of WNV transmission to humans as well. The public is often in close proximity, as well as direct contact, with both crocodilians and the vector species of WNV when attending these tourism facilities. Alligator farms open to the public may be a future cause for concern about WNV transmission in the public health sector. Another confounding factor to the human–wildlife interface is environmental modifications that promote vector abundance. For instance, agriculture and urban green space inadvertently provide habitats for mosquito vectors while providing important human services [2,49,50]. Further, WNV has a highly complex emergence tied not only to host and vector species but also to weather patterns and fine-scale landscapes [51,52]. Future studies can attempt to predict geographic patterns through the integrated modeling of these factors on smaller regional scales.

### 4.4. Avian Influence

While the overlap between crocodilians and mosquito vectors is the focus of this study, the influence of avian species as hosts is vital to understanding the continuing emergence of WNV. Birds remain the primary hosts for WNV in the environment, with host competency varying by species [15]. Recognizing the contribution that these avian host species have in the ecology of WNV is key to estimating cross-species disease transmission and how this transmission will experience a geographic shift over time.

American robins (*Turdus migratorius*), which share much of their geographic range with *A. mississippiensis*, are a notable amplifying host of WNV [53] with relatively high host competency. This is a cause for concern, as *T. migratorius* is a conspicuous species of ‘backyard bird’, often spotted around human communities. Other ‘backyard bird’ species with high WNV host competency within the range of *A. mississippiensis* include the house finch (*Haemorhous mexicanus*), the house sparrow (*Passer domesticus*), the song sparrow (*Melospiza melodia*), and the common grackle (*Quiscalus quiscula*) [15].

Species with low host competency, known as super suppressor hosts, slow virus transmission throughout local avian and mosquito populations [54]. The Northern cardinal (*Cardinalis cardinalis*), gray catbird (*Dumetella carolinensis*), and Northern mockingbird (*Mimus polyglottos*) all represent super suppressor species that can be found in the geographic range of *A. mississippiensis*. Furthermore, both sets of these super spreader and super suppressor avian species overlap to some extent across their ranges with all three *Culex* species.

### 4.5. Other Considerations

An increased diversity of mosquito species decreases the nuisance pressure and the WNV capability of *Culex* mosquitoes through the dilution effect [43]. This was found to be particularly important with *Cx. pipiens*. Therefore, we predict that if all three species of *Culex* co-occur with any of the four crocodilian species, and if the overall diversity of mosquitoes is low, the likelihood of transmission of WNV will increase the creation of a potential outbreak hotspot. Our results suggest that this scenario will be more likely for *A. mississippiensis*, as there is a lower mosquito richness across its range outside of Florida. This coincides with a high number of alligator farms (Figure 5). Finally, high levels of mosquito diversity in Central America (Figure 5) may also explain why *C. crocodilus*, which has been found in close proximity to antibody-positive horses, did not have WNV antibodies [55].

## 5. Conclusions

This study provides human and animal health reasons for a reduction in high-density alligator farming practices. In the most extreme case, environmental transmission through water has been tested and proven possible in a laboratory setting [27]. This along with the fact that WNV viremia can persist in alligators ≥ 14 days [28] makes them a particularly concerning reservoir for the persistence of WNV in the environment. The high density of alligator farms in the US will likely be most affected by WNV. The species’ distribution and the location of alligator farms in areas with all three *Culex* species and the low richness of other mosquito species make the perfect storm for WNV with the potential for transmission to humans.

## Figures and Tables

**Figure 1 microorganisms-12-01898-f001:**
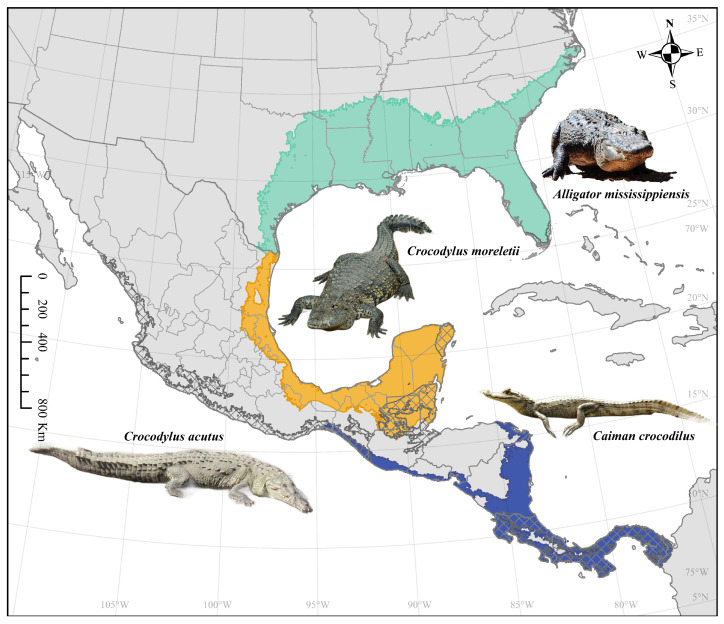
Crocodilian distributions in North America. *Alligator mississippiensis* is represented by teal, *Caiman crocodilus* by dark blue, *Crocodylus acutus* by cross-hatching, and *Crocodylus moreletii* by orange. Image credits: Desiree Andersen (*Alligator mississippiensis*, *Crocodylus acutus*), Gautier Poupeau (*Crocodylus moreletii*; Creative Commons Attribution-Share Alike 3.0 Unported license; https://commons.wikimedia.org/wiki/File:Crocodile_de_Morelet.jpeg, accessed on 4 August 2024), and Gail Hampshire (*Caiman crocodylus*; Creative Commons Attribution 2.0 Generic license; https://commons.wikimedia.org/wiki/File:Caiman_crocodilus._Spectacled_Caiman_%2842253684125%29.jpg, accessed on 4 August 2024).

**Figure 2 microorganisms-12-01898-f002:**
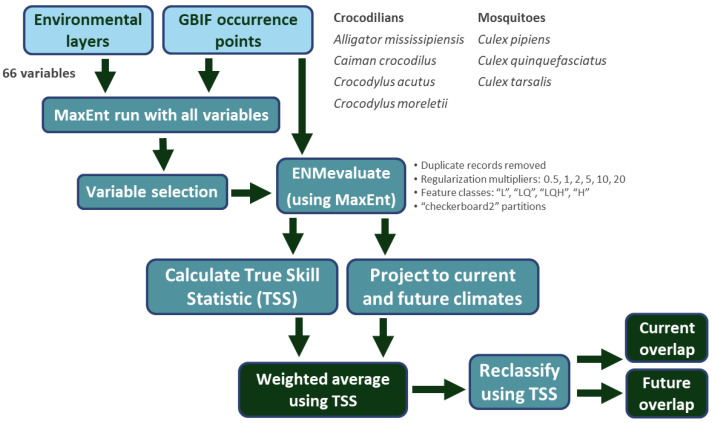
Modeling flow and components of this study.

**Figure 3 microorganisms-12-01898-f003:**
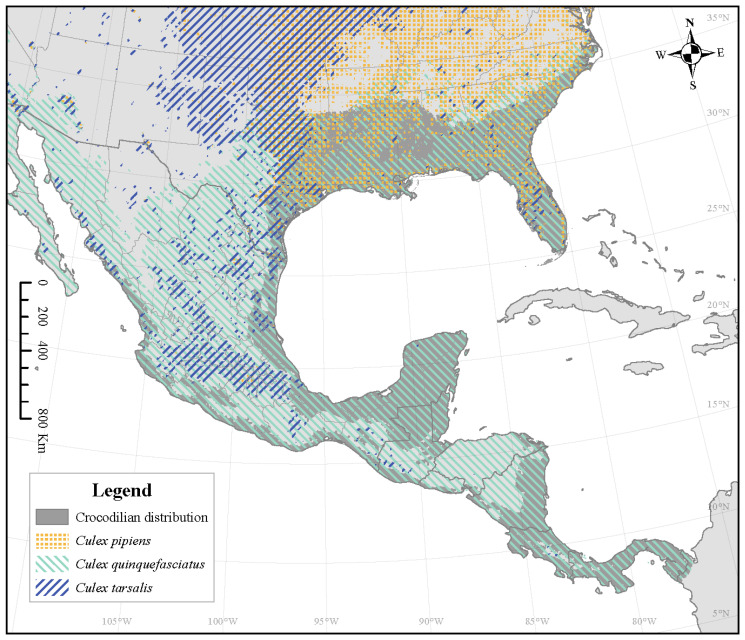
Crocodilian distribution (dark gray) overlaid with three West Nile Virus vector Culex mosquito species’ distributions, representing the greatest overlap for all climate change projections.

**Figure 4 microorganisms-12-01898-f004:**
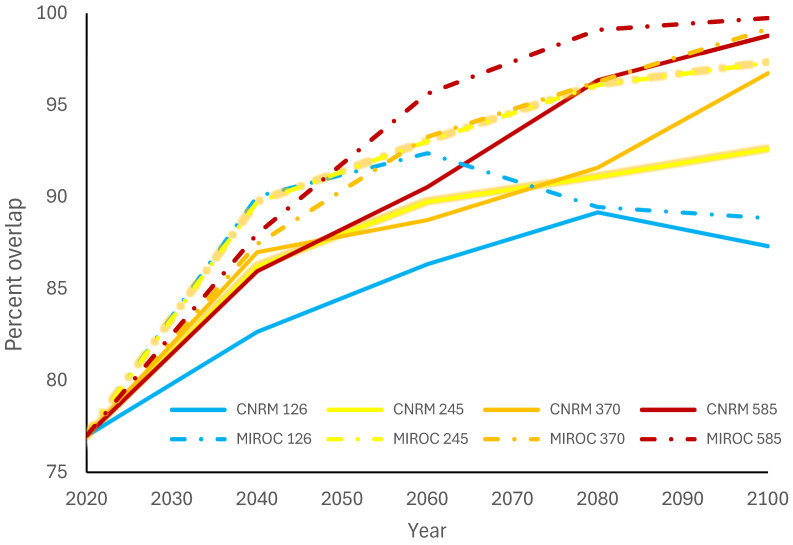
Percent of *Alligator mississippiensis* distribution overlapped with *Culex quiquefasciatus* through 2100 for four Shared Socioeconomic Pathways (climate change scenarios) and two climate models.

**Figure 5 microorganisms-12-01898-f005:**
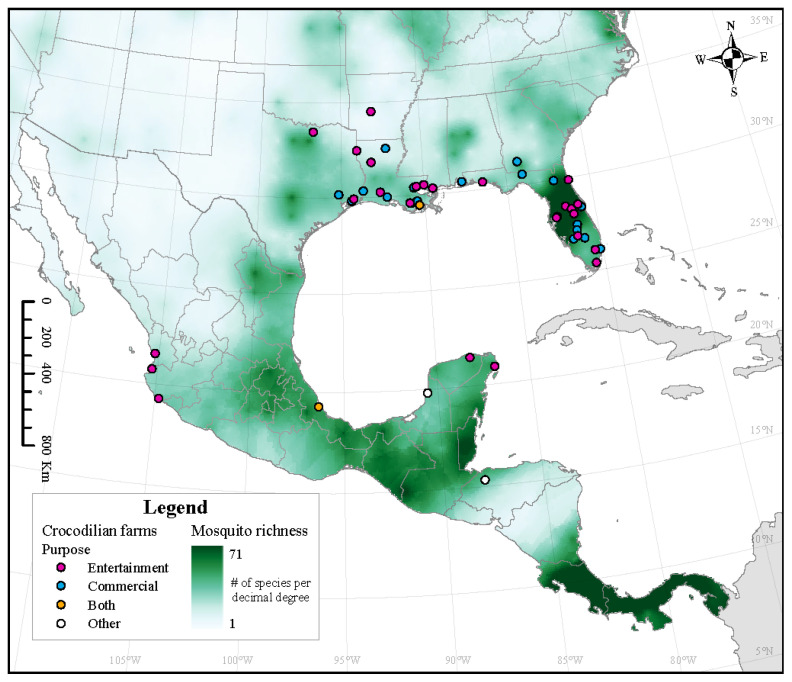
Estimated mosquito richness across North America overlaid with locations of crocodilian farms. Crocodilian farms are categorized as entertainment (or tourism), commercial (hide/meat/hunt), both (entertainment and commercial), or other. Crocodilian farms are listed in Appendix A.

**Table 1 microorganisms-12-01898-t001:** Overlap percent of Culex species with crocodilian species and the low and high predictions for all climate change scenarios. The scenarios represented by percent increase or decrease are notated in parentheses with the climate model (CNRM or MIROC), Shared Socioeconomic Pathway, and year.

	Current	Lowest Prediction	Highest Prediction
*Culex pipiens*			
* Alligator mississippiensis*	63.2%	−4.9% (MIROC 585 2100)	−0.9% (CNRM 126 2040)
* Caiman crocodilus*	0.2%	−0.1% or less (all scenarios)	
* Crocodylus acutus*	1.1%	−0.2% (BOTH 585 2100)	−0.1% (multiple)
* Crocodylus moreletii*	4.7%	−4.1% (MROC 585 2100)	−2.5% (CNRM 126 2040)
*Culex quinquefasciatus*			
* Alligator mississippiensis*	77.0%	**+5.7% (CNRM 126 2040)**	**+22.7% (MIROC 585 2100)**
* Caiman crocodilus*	98.4%	No change	
* Crocodylus acutus*	98.5%	No change	
* Crocodylus moreletii*	99.5%	No change	
*Culex tarsalis*			
* Alligator mississippiensis*	32.3%	−4.5% (MIROC 585 2100)	**+4.0% (CNRM 585 2080)**
* Caiman crocodilus*	8.6%	−0.6% (MIROC 585 2100)	−2.0% (MIROC 245 2060)
* Crocodylus acutus*	16.6%	−5.4% (CNRM 585 2100)	−1.3% (MIROC 370 2040)
* Crocodylus moreletii*	20.2%	−8.0% (MIROC 585 2100)	−2.4% (MIROC 585 2040)

Increases in overlap are bolded.

## Data Availability

The data are contained within the article or Appendix A.

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
