# Peer review of "The Alligator and the Mosquito: North American Crocodilians as Amplifiers of West Nile Virus in Changing Climates"

_microorganisms, 2024, doi:10.3390/microorganisms12091898_

Round 1

Reviewer 1 Report

Comments and Suggestions for Authors

I have reviewed the manuscript titled “The alligator and the mosquito: North American crocodilians as amplifiers of West Nile Virus in changing climates” by Andersen et al. 

This manuscript present an interesting analysis of the distribution change in crocodilians and mosquitoes (vectors of WNV) with anticipated climate change. There is some clear evidence that WNV activity has occurred in association with alligator/crocodile farming operations and considering future risk with differing mosquito abundance and distribution, as well as non-farmed populations of these animals is important.

It may be worth a statement from authors regarding the role of “wild” and “farmed” crocodilians in WNV transmission factors. While I would assume that most studies centre on farmed populations, a specific comment on wildlife populations would be useful in introduction.

While the authors have been though in their analysis of crocodilians and mosquitoes, I feel there is strong evidence that the future activity of WNV, even in areas where these crocodilians (farmed or non-farmed) are present will be determined by avian activity. As the primary amplifying hosts of WNV, their presence will remain of critical importance. With this in mind, it may be useful to include some comments/discussion regarding how bird activity in and around farming/tourist operations may also influence spill over of WNV. A detailed modelling exercise is not necessary but some additional commentary would be beneficial.

Beyond the environmental/climatic factors investigated to determine future changes in the future mosquito populations, consideration may be most important to the landscape immediately within and around farming operations. While the models are predicting suitability on a region scale, are there issues in and around farming operations that increase the risk of abundance mosquito populations more substantially. For example, the close association of Culex quinquefasciatus with urban stormwater/wastewater infrastructure may elevate the local risk for some farming operations. Some of these issues could be dealt with in an additional section of manuscript dealing briefly with strategies to manage future risk to farms (and associated human activity) at risk of WNV.

The authors may wish to consider some work done on crocodiles in Australia and their susceptibility to and role in transmission of Kunjin virus (a subtype of WNV). There has been a number of studies focused on this situation and I encourage the authors to review and see if any additional perspectives may be offered for consideration in their manuscript.

Some examples of papers include:

Habarugira, G., Moran, J., Colmant, A.M., Davis, S.S., O’Brien, C.A., Hall-Mendelin, S., McMahon, J., Hewitson, G., Nair, N., Barcelon, J. and Suen, W.W., 2020. Mosquito-independent transmission of West Nile virus in farmed saltwater crocodiles (Crocodylus porosus). Viruses, 12(2), p.198.

Isberg, S.R., Moran, J.L., De Araujo, R., Elliott, N., Davis, S.S. and Melville, L., 2019. First evidence of Kunjin strain of West Nile virus associated with saltwater crocodile (Crocodylus porosus) skin lesions. Australian veterinary journal, 97(10), pp.390-393.

Kurucz, N., McMahon, J.L., Warchot, A., Hewitson, G., Barcelon, J., Moore, F., Moran, J., Harrison, J.J., Colmant, A.M., Staunton, K.M. and Ritchie, S.A., 2022. Nucleic acid preservation card surveillance is effective for monitoring arbovirus transmission on crocodile farms and provides a One Health benefit to northern Australia. Viruses, 14(6), p.1342.

Some minor comments.

The manuscript should be reviewed with particular attention to scientific names and their abbreviations (crocodilians and mosquitoes). There is some inconsistencies in the use of abbreviations after first use (noting that full scientific names are used to start sentences). It is also important to consider that the correct abbreviation of mosquito genera of Culex is Cx., Aedes is Ae. etc and manuscript should be checked for corrections.

Author Response

Comment 1: I have reviewed the manuscript titled “The alligator and the mosquito: North American crocodilians as amplifiers of West Nile Virus in changing climates” by Andersen et al. 

This manuscript present an interesting analysis of the distribution change in crocodilians and mosquitoes (vectors of WNV) with anticipated climate change. There is some clear evidence that WNV activity has occurred in association with alligator/crocodile farming operations and considering future risk with differing mosquito abundance and distribution, as well as non-farmed populations of these animals is important.

Response 1: Thank you for your comments and suggestions.

Comment 2: It may be worth a statement from authors regarding the role of “wild” and “farmed” crocodilians in WNV transmission factors. While I would assume that most studies centre on farmed populations, a specific comment on wildlife populations would be useful in introduction.

Response 2: We have already included a statement to this effect in the introduction: ”Incidence of WNV is however not limited to American alligators; WNV antibodies haves also been found in wild and farmed Morelet’s, or Mexican crocodiles, Crocodylus moreletii, indicating susceptibility of crocodilians as a group to WNV.”

We have also added the following pertaining to farmed and natural densities of alligators: 

”Natural densities of alligators range depending on local conditions from 1.52-2.35 alligators/km in a wildlife area in east Texas [22] to around 2.56-9.02 alligators/km in Florida [23]” (Line 51-53)

“Recommended stocking rates for farmed alligators are around 10-20 alligators/acre, however, densities can be as high as 50/acre at some farming locations [24,25]” (Line 55-57)

Additionally, we have added more detail to the same paragraph (Line 50-73) pertaining to effects and transmission dynamics of different species in varying contexts.

Comment 3: While the authors have been though in their analysis of crocodilians and mosquitoes, I feel there is strong evidence that the future activity of WNV, even in areas where these crocodilians (farmed or non-farmed) are present will be determined by avian activity. As the primary amplifying hosts of WNV, their presence will remain of critical importance. With this in mind, it may be useful to include some comments/discussion regarding how bird activity in and around farming/tourist operations may also influence spill over of WNV. A detailed modelling exercise is not necessary but some additional commentary would be beneficial.

Response 3: As suggested, we have added a section (lines 284-304) to our discussion on avian species overlap: 

4.4. Avian influence

While the overlap between crocodilians and mosquito vectors is the focus of this study, the influence of avian species as hosts is vital to understanding the continuing emergence of WNV. Birds remain the primary hosts for WNV in the environment, with host competency varying by species [15]. Recognizing the contribution that these avian host species have in the ecology of WNV is key to estimating cross-species disease transmission and how this transmission will experience a geographic shift over time.

American robins (Turdus migratorius), which share much of their geographic range with A. mississippiensis, are a notable amplifying host of WNV [53] with relatively high host competency. This is a cause for concern, as T. migratorius is a conspicuous species of ‘backyard bird’, often spotted around human communities. Other ‘backyard bird’ species with high WNV host competency within the range of A. mississippiensis include the house finch (Haemorhous mexicanus), the house sparrow (Passer domesticus), the song sparrow (Melospiza melodia), and the common grackle (Quiscalus quiscula) [15].

Species with low host competency, known as super suppressor hosts, slow virus transmission throughout local avian and mosquito populations [54]. The northern cardinal (Cardinalis cardinalis), gray catbird (Dumetella carolinensis), and northern mockingbird (Mimus polyglottos) all represent super suppressor species that can be found in the geographic range of A. mississippiensis. Furthermore, both sets of these super spreader and super suppressor avian species overlap to some extent across their ranges with all three Culex species.

Comment 4: Beyond the environmental/climatic factors investigated to determine future changes in the future mosquito populations, consideration may be most important to the landscape immediately within and around farming operations. While the models are predicting suitability on a region scale, are there issues in and around farming operations that increase the risk of abundance mosquito populations more substantially. For example, the close association of Culex quinquefasciatus with urban stormwater/wastewater infrastructure may elevate the local risk for some farming operations. Some of these issues could be dealt with in an additional section of manuscript dealing briefly with strategies to manage future risk to farms (and associated human activity) at risk of WNV.

Response 4: As suggested, we have included the following (lines 277-283) "Another confounding factor to the human-wildlife interface is environmental modification that promotes vector abundance. For instance, agriculture and urban greenspace inadvertently provide habitat for mosquito vectors while providing important human services [2,49,50]. Further, WNV has highly complex emergence tied not only to host and vector species, but also to weather patterns and fine-scale landscape [51,52]. Future studies can attempt to predict geographic patterns through integrated modeling of these factors on smaller regional scales."

Comment 5:  The authors may wish to consider some work done on crocodiles in Australia and their susceptibility to and role in transmission of Kunjin virus (a subtype of WNV). There has been a number of studies focused on this situation and I encourage the authors to review and see if any additional perspectives may be offered for consideration in their manuscript.

Some examples of papers include:

Habarugira, G., Moran, J., Colmant, A.M., Davis, S.S., O’Brien, C.A., Hall-Mendelin, S., McMahon, J., Hewitson, G., Nair, N., Barcelon, J. and Suen, W.W., 2020. Mosquito-independent transmission of West Nile virus in farmed saltwater crocodiles (Crocodylus porosus). Viruses, 12(2), p.198.

Isberg, S.R., Moran, J.L., De Araujo, R., Elliott, N., Davis, S.S. and Melville, L., 2019. First evidence of Kunjin strain of West Nile virus associated with saltwater crocodile (Crocodylus porosus) skin lesions. Australian veterinary journal, 97(10), pp.390-393.

Kurucz, N., McMahon, J.L., Warchot, A., Hewitson, G., Barcelon, J., Moore, F., Moran, J., Harrison, J.J., Colmant, A.M., Staunton, K.M. and Ritchie, S.A., 2022. Nucleic acid preservation card surveillance is effective for monitoring arbovirus transmission on crocodile farms and provides a One Health benefit to northern Australia. Viruses, 14(6), p.1342.

Response 5: Thank you for providing these insights. We have included the following lines in the introduction (lines 68-73) "Outside of North America, the Kunjin strain of WNV has been detected in farmed saltwater crocodiles in Australia, where it may be the cause of skin lesions [30]. Further studies in farmed saltwater crocodiles in Northern Australia show this strain is capable of mosquito-independent transmission, meaning the crocodile is capable of amplification and transmission through water via cloacal shedding [31]." 

Comment 6: Some minor comments. The manuscript should be reviewed with particular attention to scientific names and their abbreviations (crocodilians and mosquitoes). There is some inconsistencies in the use of abbreviations after first use (noting that full scientific names are used to start sentences). It is also important to consider that the correct abbreviation of mosquito genera of Culex is Cx., Aedes is Ae. etc and manuscript should be checked for corrections.

Response 6: Thank you for alerting us to these inconsistencies. We have gone through and checked that scientific names and abbreviations are consistent throughout and follow proper conventions.

Reviewer 2 Report

Comments and Suggestions for Authors

Dear Authors,

The manuscript which you submitted is very interesting and the topic can be considered of particular national interest. 

The findings obtained in this research represents a good base for the other countries with the tradition of rearing crocodilians and dealing with the West Nile virus. 

Before the manuscript is published, there are several issues that need to be adressed. 

In the abstract- it is not correct to say competent host but competent reservoir. Please correct. Same in the line 28. 

L27 The authors wrote "disease", but actually should be pathogens causing disease. 

L34 Please do not start sentence with the abbreiviation. 

L56 Word mosquitoes should not be in italics. 

L61 and 62 The latin names should be abbreiviated in the whole manuscript exept when it is at the begining of the sentence and when they are mentioned for the first time. 

There is nothing written about the Culex mosquitoes' biology. That is very relevant for this study. Please add it. 

You did not mention anything about infection of crocodilians. What are the symptoms? What type of WNV  manifestation has been reported for these animals so far?

It should be added connection of mosquitoes and crocodilians. How high is the likelyhood that these three species with blood-feed on the crocodilians?

Results:

Please abbreviate correctly Culex in the whole manuscript. You wrote C. pipiens, but it is actually Cx. pipiens. 

It is important to point out where did the mosquitoes sampled and how. I suppose you could provide that from the database which you use to obtain the mosquito monitoring results. 

L214 It is given pipiens and pipiens. Is that mistake?

L265 High should be in small letter. 

L265-268 This should be modified because the WNV is very complex disease and it is still not completely clear what is the reason of the disease "explosions" in some years in human population. Therefore, this sentence cannot be accepted in this form. 

Author Response

Comment 1: Dear Authors, The manuscript which you submitted is very interesting and the topic can be considered of particular national interest. The findings obtained in this research represents a good base for the other countries with the tradition of rearing crocodilians and dealing with the West Nile virus. 

Response 1: Thank you for your comments and suggestions.

Comment 2: In the abstract- it is not correct to say competent host but competent reservoir. Please correct. Same in the line 28. 

Response 2: While we acknowledge the term “reservoir” may be more appropriate in some contexts, the term “reservoir” refers to a subset of hosts that does not experience symptoms. As American alligators can experience symptoms including mortality, we have chosen to keep the term “host” for the context of our study.

Comment 3: L27 The authors wrote "disease", but actually should be pathogens causing disease. 

Response 3: We have added the term “pathogen” here. (Line 27)

Comment 4: L34 Please do not start sentence with the abbreiviation. 

Response 4: We have revised so the sentence no longer starts with an abbreviation. (Line 34)

Comment 5: L56 Word mosquitoes should not be in italics. 

Response 5: The referenced word is no longer in italics. (Line 60)

Comment 6: L61 and 62 The latin names should be abbreiviated in the whole manuscript exept when it is at the begining of the sentence and when they are mentioned for the first time. 

Response 6: We have gone through and checked that scientific names and abbreviations are consistent throughout and follow proper conventions.

Comment 7: There is nothing written about the Culex mosquitoes' biology. That is very relevant for this study. Please add it. 

Response 7: We agree that the influence of ecology on mosquito biology and abundance is relevant to this study, and we have added the following to the discussion: “Another confounding factor to the human-wildlife interface is environmental modification that promotes vector abundance. For instance, agriculture and urban greenspace inadvertently provide habitat for mos-quito vectors while providing important human services [2,49,50].” (Line 277-280)

Comment 8: You did not mention anything about infection of crocodilians. What are the symptoms? What type of WNV  manifestation has been reported for these animals so far?

Response 8: To address this comment, we have revised part of the introduction to include details on infection and symptoms of crocodilians in varying contexts: “In addition to causing lesions which reduce the economic value of alligator hides [29], WNV can cause large, fatal outbreaks in juveniles [28]. Incidence of WNV is however not limited to American alligators; WNV antibodies have also been found in wild and farmed Morelet’s, or Mexican crocodiles, Crocodylus moreletii, indicating susceptibility of crocodilians as a group to WNV, although the species’ ability to amplify the virus and transmit it back to mosquitos has not been tested. Outside of North America, the Kunjin strain of WNV has been detected in farmed saltwater crocodiles in Australia, where it may be the cause of skin lesions [30]. Further studies in farmed saltwater crocodiles in Northern Australia show this strain is capable of mosquito-independent transmission, meaning the crocodile is capable of amplification and transmission through water via cloacal shedding [31].” (Line 63-73)

Comment 9: It should be added connection of mosquitoes and crocodilians. How high is the likelyhood that these three species with blood-feed on the crocodilians?

Response 9: We have added a sentence about a previous study by Byas et al., 2022 on the rates of blood feeding by Culex species on alligators: “Of the Culex species tested for propensity to feed on A. mississippiensis, feeding by Cx. quinquefasciatus was highest at 82.2%, followed by Cx. pipiens at 31.1%, and Cx. tarsalis at 6.7% [27].” (Line 61-63)

Results:

Comment 10: Please abbreviate correctly Culex in the whole manuscript. You wrote C. pipiens, but it is actually Cx. pipiens. 

Response 10: We have gone through and checked that scientific names and abbreviations are consistent throughout and follow proper conventions.

Comment 11: It is important to point out where did the mosquitoes sampled and how. I suppose you could provide that from the database which you use to obtain the mosquito monitoring results. 

Response 11: Occurrence data was collected from GBIF, an open-source database of occurrence data. We did not conduct sampling ourselves.

Comment 12: L214 It is given pipiens and pipiens. Is that mistake?

Response 12: We have corrected to “quinquefasciatus”. (Line 235)

Comment 13: L265 High should be in small letter. 

Response 13: We have corrected as suggested. (Line 313)

Comment 14: L265-268 This should be modified because the WNV is very complex disease and it is still not completely clear what is the reason of the disease "explosions" in some years in human population. Therefore, this sentence cannot be accepted in this form. 

Response 14: We have kept the referenced sentence as it contains context as to why Caimans may not experience impacts of WNV. However, we have added the following to address the complexity of the virus’ emergence: “Further, WNV has highly complex emergence tied not only to host and vector species, but also to weather patterns and fine-scale landscape [47,48]. Future studies can attempt to predict geographic patterns through integrated modeling of these factors on smaller regional scales.” (Line 280-283)

Reviewer 3 Report

Comments and Suggestions for Authors

This is a theoretical analysis using maximum entropy models to map current and future distributions of three mosquito vector species and four crocodilian species in North America with the goal of identifying the “emerging” risk of WNV outbreaks associated with changing climates and West Nile Virus (WNV) associated with crocodilians in North America. There are similar studies undertaken with avian species and mosquitoes, and comparison of these studies with the current study may help the analysis.  

The authors make the important point that the mosquito species currently with the highest overlap for all crocodilian species is Culex quinquefasciatus, which covers 77.0% A. mississippiensis’ range. How does this range compare to (1) distribution of particular avian species associated with WNV transmission and more importantly (2) historical human outbreaks of WN disease? Such a comparison would improve the Discussion.

Introduction: I would suggest some text on whether or not each alligator species can transmit WNV to mosquitoes or are dead end hosts would be helpful. As written, this is not clear.

Author Response

Comment 1: This is a theoretical analysis using maximum entropy models to map current and future distributions of three mosquito vector species and four crocodilian species in North America with the goal of identifying the “emerging” risk of WNV outbreaks associated with changing climates and West Nile Virus (WNV) associated with crocodilians in North America. There are similar studies undertaken with avian species and mosquitoes, and comparison of these studies with the current study may help the analysis.  

Response 1: Thank you for your comments and suggestions. As suggested, we have added a section to the discussion on avian species. (Line 284-304)

Comment 2: The authors make the important point that the mosquito species currently with the highest overlap for all crocodilian species is Culex quinquefasciatus, which covers 77.0% A. mississippiensis’ range. How does this range compare to (1) distribution of particular avian species associated with WNV transmission and more importantly (2) historical human outbreaks of WN disease? Such a comparison would improve the Discussion.

Response 2: We have added a section on avian species and their distributions to the discussion. (Line 284-304)

Comment 3: Introduction: I would suggest some text on whether or not each alligator species can transmit WNV to mosquitoes or are dead end hosts would be helpful. As written, this is not clear.

Response 3: In the original text, we state: “However, WNV has been found in such farmed alligators at amplified levels sufficient for transmission to humans via Culex mosquitoes [27,28]” (Line 58-60) on the ability of alligators to amplify and therefore be able to transmit the virus. 

We have also added the following to clarify that this has not been tested in the crocodilian species Crocodylus moreletii: “although the species’ ability to amplify the virus and transmit it back to mosquitos has not been tested” (Line 67-68).

Additionally, we have added the following on each mosquito species’ propensity to feed on Alligator mississippiensis when tested: “Of the Culex species tested for propensity to feed on A. mississippiensis, feeding by Cx. quinquefasciatus was highest at 82.2%, followed by Cx. pipiens at 31.1%, and Cx. tarsalis at 6.7% [27].” (Line 61-63)